# Multi-Omics Characterization of Inflammatory Bowel Disease-Induced Hyperplasia/Dysplasia in the *Rag2^−/−^*/*Il10^−/−^* Mouse Model

**DOI:** 10.3390/ijms22010364

**Published:** 2020-12-31

**Authors:** Qiyuan Han, Thomas J. Y. Kono, Charles G. Knutson, Nicola M. Parry, Christopher L. Seiler, James G. Fox, Steven R. Tannenbaum, Natalia Y. Tretyakova

**Affiliations:** 1Department of Biochemistry, Molecular Biology and Biophysics, University of Minnesota, Minneapolis, MN 55455, USA; hanxx963@umn.edu; 2Minnesota Supercomputing Institute, University of Minnesota, Minneapolis, MN 55455, USA; konox006@umn.edu; 3Department of Biological Engineering, Massachusetts Institute of Technology, Cambridge, MA 02139, USA; charlie.knutson@novartis.com (C.G.K.); jgfox@mit.edu (J.G.F.); srt@mit.edu (S.R.T.); 4Division of Comparative Medicine, Massachusetts Institute of Technology, Cambridge, MA 02139, USA; nicola.parry@tufts.edu; 5Department of Medicinal Chemistry, University of Minnesota, Minneapolis, MN 55455, USA; christopherlseiler@gmail.com

**Keywords:** inflammatory bowel disease, hyperplasia, dysplasia, colorectal cancer, transcriptome, methylome, hydroxymethylome, multi omics analysis and integration

## Abstract

Epigenetic dysregulation is hypothesized to play a role in the observed association between inflammatory bowel disease (IBD) and colon tumor development. In the present work, DNA methylome, hydroxymethylome, and transcriptome analyses were conducted in proximal colon tissues harvested from the *Helicobacter hepaticus *(*H. hepaticus*)**-infected murine model of IBD. Reduced representation bisulfite sequencing (RRBS) and oxidative RRBS (oxRRBS) analyses identified 1606 differentially methylated regions (DMR) and 3011 differentially hydroxymethylated regions (DhMR). These DMR/DhMR overlapped with genes that are associated with gastrointestinal disease, inflammatory disease, and cancer. RNA-seq revealed pronounced expression changes of a number of genes associated with inflammation and cancer. Several genes including *Duox2*, *Tgm2*, *Cdhr5*, and *Hk2* exhibited changes in both DNA methylation/hydroxymethylation and gene expression levels. Overall, our results suggest that chronic inflammation triggers changes in methylation and hydroxymethylation patterns in the genome, altering the expression of key tumorigenesis genes and potentially contributing to the initiation of colorectal cancer.

## 1. Introduction

The incidence of colorectal cancer (CRC) in western countries has been on the rise, now accounting for approximately 10% of total cancer related mortality [1,2]. In addition to age, diet, and changes in the microbiome, inflammatory bowel disease (IBD) such as Crohn’s disease and ulcerative colitis represent important risk factors for CRC development [3]. IBD-associated chronic inflammation triggers overproduction of reactive oxygen and nitrogen species, genotoxicity, aberrant tissue repair, and increased cell proliferation [4,5]. Furthermore, inflammation triggers dramatic changes in the epigenome that can collaborate with genetic changes to promote carcinogenesis [6,7,8,9].

129S6/SvEvTac-*Rag2*^tm1Fw^ (*Rag2*^−/−^) and 129S6/SvEvTac-*Rag2*^tm1Fw^*Il10*^−/−^(*Rag2^−/−^/Il10^−/−^*) mice infected with *H. hepaticus* were previously evaluated as a murine model of IBD [10,11,12]. While *Rag2* is important for lymphocyte development, *Il10* provides protection against inflammatory pathology induced by bacterial infection [10]. Approximately 20 weeks following infection with *H. hepaticus*, *Rag2^−/^/Il10^−/−^* mice develop severe intestinal inflammation manifested in pathological tissue changes, accumulation and infiltration of neutrophils and macrophages, increased production of reactive oxygen and nitrogen species, and oxidative DNA damage [10,11,12]. DNA isolated from intestinal tissues of infected mice shows increased levels of 5-chlorodeoxycytosine, a DNA adduct commonly used as a biomarker of inflammation [12,13]. Apart from its intrinsically mutagenic features [14], 5-chlorodeoxycytosine also promotes epigenetic changes. When found within a CpG sequence context, 5-chlorodeoxycytosine is recognized as 5-methylcytosine by maintenance DNA methyltransferase (DNMT) 1, leading to aberrant cytosine methylation [13,15,16,17]. DNA isolated from inflamed colon tissues also contains increased numbers of reactive oxygen species-induced 8-oxo-2’-deoxyguanosine adducts [18]. 8-oxo-2’-deoxyguanosine can be removed by OG-glycosylase I, yielding abasic sites [19]. If present in a potential G-quadruplex-forming sequence of promoter, abasic sites can trigger a transition from B DNA to a G-quadruplex structure. This in turn can lead to gene activation via recruitment of purinic/apyrimidinic endonuclease 1 and transcription factors to abasic sites [20]. *H. hepaticus*-infected *Rag2^−/−^/Il10^−/−^* mice develop colon carcinomas by 4 to 8 months post infection [10,21].

5-Methylcytosine (5mC) is by far the most abundant epigenetic mark of DNA (3–5% of all cytosines). In mammals, 5mC is formed via DNA methyltransferase-mediated methylation of the C-5 carbon on cytosine residues in the context of CpG dinucleotides [22]. In eukaryotes, DNMT 3a/b introduce de novo methylation marks in DNA during development, while tissue-specific cytosine methylation patterns are maintained during replication through the activity of DNMT1 [23,24]. As part of epigenetic regulation, the presence of 5mC in promoter sequences is associated with gene silencing [25] while the removal of methylation marks leads to gene reactivation [26].

Ten-eleven translocation dioxygenases (TET1-3) convert 5mC in DNA to 5-hydroxymethyl-cytosine (5hmC), 5-formylcytosine (5fC), and 5-carboxylcytosine (5caC) [27]. Both 5fC and 5caC can be recognized by the base excision repair machinery and can be replaced with cytosine, leading to active DNA demethylation [28,29]. Additionally, oxidized forms of 5mC may act as epigenetic marks in their own right by interacting with a distinct set of protein “readers” [30]. Genome-wide profiling of 5hmC during in vitro colonic differentiation revealed that 5hmC residues are overrepresented in both promoter and gene body regions of highly expressed genes and may be associated with intestinal transcription factor binding [31]. 5fC is preferentially deposited at poised enhancer regions [32], and generation of 5fC at promoter region proceeds the upregulation of gene expression in mouse early embryos [33]. 5caC tends to asymmetrically distribute on the antisense strand of highly expressed genes coding for proteins [34].

In healthy cells and tissues, the balance of DNA methylation and demethylation is maintained by the opposing actions of DNMT and TET enzymes. However, this equilibrium is disrupted in cancer. Significant changes in DNA methylation patterns such as global hypomethylation, promoter hypermethylation, and decreased 5hmC levels are observed in many tumor types including CRC [35,36,37,38,39,40,41]. Previous studies employed bisulfite sequencing in an attempt to map DNA methylation changes in CRC [42,43]. In this approach, DNA is treated with sodium bisulfite to convert all cytosine bases to uracil, which is read as thymine during further amplification and sequencing steps. 5mC bases resist bisulfite-induced deamination and behave similarly to unmodified cytosine in downstream amplification and sequencing, making it possible to identify the 5mC sites by comparing bisulfite sequencing results to a reference genome [44]. Unfortunately, this method cannot distinguish between 5mC and 5hmC as both bases resist deamination during sodium bisulfite treatment, while unsubstituted cytosine, 5fC, and 5caC undergo deamination and are read as T [45,46]. Therefore, previous investigations failed to separately map these two epigenetic marks, limiting our understanding of fundamental epigenetic mechanisms in CRC.

The present study employed an updated version of reduced representation bisulfite sequencing (RRBS + oxRRBS) [46] to separately map IBD-induced changes of DNA hydroxymethylome and methylome across the genome in the *Rag2^−/−^*/*Il10^−/−^ H. hepaticus*-infected murine model of IBD. Combined with global gene expression analyses via RNA-seq, our results provide the first comprehensive view of epigenetic alterations induced by inflammation in the colon, yielding useful information for further research on IBD biomarker discovery and potentially facilitating future development of new therapies for IBD.

## 2. Results

### 2.1. Histopathology Analysis of Colon Tissues from H. hepaticus-Infected Rag2^−/−^/Il10^−/−^Mice

*Rag**2*^−/−^/*Il**10*^−/−^ mice [10] were used as an animal model of IBD. These mice lack functional lymphocytes due to a deficiency in the recombinase-activating gene-2 gene (*Rag*2). Since lymphocytes prevent infection-induced inflammation and block the formation of primary tumors of intestinal epithelia in the bowel, these mice have a markedly higher frequency of lower bowel carcinoma as compared with congenic wildtype mice [10]. Additional IL10**** deficiency in this model imparts an increased sensitivity to inflammation because anti-inflammatory activities of IL10 can inhibit TNFα-dependent responses to pathogenic bacteria [10,21,47,48].

*Rag**2*^−/−^/*Il**10*^−/−^ mice were inoculated with *H. hepaticus* at 6–7 weeks of age (Figure 1A). *H. hepaticus* can colonize the ceca and colons of mice and can influence pathological outcomes along the lower bowel [49]. To characterize disease status in the *Rag**2*^−/−^/*Il**10*^−/−^ mouse model of IBD, colon tissues (from cecum to distal colon) were harvested from infected and control mice 20 weeks post infection (27 weeks of age). Histopathological results revealed the development of inflammation, hyperplasia, and dysplasia along the lower bowel of treated mice but not in controls (Figure 1B and Appendix A). Histological activity indices of colitis (cumulative score of inflammation, edema, epithelial defects, crypt atrophy, hyperplasia, and dysplasia) in each part of the colon (ileocal junction, cecal greater curvature, proximal colon, transverse colon, and distal colon) were significantly higher in *H. hepaticus*-infected mice as compared to controls (Figure 1B). These results are consistent with previous observations [12], confirming that *H. hepaticus* infection resulted in pronounced inflammation, cell transformation, and morphological changes along the lower bowel of *Rag**2*^−/−^/*Il**10*^−/−^ mice.

### 2.2. Characterization of Methylome and Hydroxymethylome in IBD-Induced Hyperplasia/Dysplasia

In order to detect global cytosine methylation and hydroxymethylation changes associated with IBD, an isotope dilution HPLC-ESI^+^-MS/MS assay developed in our laboratory [50] was employed to quantify total 5mC and 5hmC levels in genomic DNA isolated from various sections of the gastrointestinal tract of control and IBD mice (Figure 1C,D). We found that the overall DNA methylation and hydroxymethylation levels were comparable across different parts of the colon. IBD was associated with a significant decrease in DNA methylation in the transverse colon (3.55 ± 0.30% vs. 3.01 ± 0.12%, *p* < 0.03) and distal colon (3.61 ± 0.16% vs. 3.17 ± 0.14%, *p* < 0.00008). DNA isolated from cecum and proximal colon tissues of IBD mice had significantly lower global cytosine hydroxymethylation levels (up to 43% decrease) compared to controls (Figure 1D). The observed differences in global 5mC and 5hmC alterations across the gastrointestinal tract may reflect spatial differences in the extent and the pathways of chronic inflammation.

Although the observed global changes in 5mC and 5hmC amounts in colonic tissues of IBD mice (Figure 1C,D) are indicative of epigenetic deregulation, full understanding of their biological significance requires mapping epigenetic changes to specific sites and regions within the genome. Proximal colon tissues were selected for these analyses based on our histopathology results (Figure 1B) and the significant global change of 5hmC, which is a hallmark of many cancer types including CRC [51] (Figure 1C,D). Pathology scores for these mice are summarized in Figure 1B and Appendix A (Appendix A).

In order to map IBD-induced changes in cytosine methylation and hydroxymethylation across the genome, genomic DNA was extracted from proximal colon tissues of IBD and control mice (3 per group). A combination of reduced representation bisulfite sequencing (RRBS) and oxidative-reduced representation bisulfite sequencing (oxRRBS) [46] was used to examine genome-wide patterns of methylation and hydroxymethylation. In this methodology, genomic DNA is first treated with the MspI enzyme, which specifically recognizes and cuts CCGG sites of DNA. This enzymatic digestion step enables enrichment of CpG sites for the downstream sequencing. The digested DNA is subjected to traditional bisulfite treatment and oxidation-bisulfite treatment, respectively. In the oxidation-bisulfite treatment, we employed potassium perruthenate (KRuO_4_) to oxidize 5hmC to 5fC, therefore yielding an accurate 5mC fraction at each cytosine when compared to the reference genome. By comparing the oxRRBS dataset with RRBS datasets, an accurate 5hmC fraction can be calculated (Figure 2A). Previously, we successfully employed this approach to characterize methylation and hydroxymethylation changes in an animal model of lung cancer (A/J mice treated with lipopolysaccharide) [52].

RRBS analysis of genomic DNA isolated from proximal colon tissues of *H. hepaticus*-infected and control *Rag*2^−/−^/*Il**10*^−/−^ mice resulted in 23,447,868 (± 5,076,283) read pairs for each sample. oxRRBS yielded a mean of 29,890,775 (± 2,254,572) read pairs for each sample. As CpG is the primary sequence context for cytosine methylation and both strands of DNA are typically methylated, we have focused our analysis on symmetrically methylated an hydroxymethylated sites. After filtering for symmetric methylation, blacklisted regions, and sequencing depth, a total of 1,048,884 CpG sites were covered by both RRBS and oxRRBS datasets. Multidimensional scaling analysis revealed a clear separation between IBD and control groups based on methylation patterns (Appendix A), although no clear separation was found based on the hydroxymethylation patterns (Appendix A).

### 2.3. Differential Methylation Analysis Reveals Extensive DMRs and DhMRs in IBD-Induced Hyperplasia/Dysplasia

Methylome and hydoxymethylome characterization by RRBS/oxRRBS revealed that IBD-induced inflammation in proximal colon caused significant alterations of the deposition patterns of both DNA epigenetic marks (5mC and 5hmC). DNA isolated from the IBD group was characterized by a total of 1606 differentially methylated regions (DMRs) and 3011 differentially hydroxymethylated regions (DhMRs). DMR/DhMR were defined as genomic regions that had at least 3 CpG sites within a 200-bp genomic window with a false discovery rate at less than 0.05. Although the total number of DhMRs was higher than that of DMRs, methylation differences were more extensive than changes in cytosine hydroxymethylation (Figure 2B,C). Of all DMRs, we identified 784 regions with decreased methylation and 822 regions with increased methylation. Among DhMRs, 1454 regions had decreased hydroxymethylation and 1557 regions had increased hydroxymethylation. Ingenuity Pathway Analysis (IPA) revealed that DMRs were preferentially found in genes associated with cancer, cell death and survival, gastrointestinal disease, organismal injury and abnormities, inflammatory disease, and inflammatory response (Figure 2D) while DhMRs were found in genes associated with digestive system development and function, organ and tissue morphology, gastrointestinal disease, and inflammatory disease (Figure 2E). These results suggested a potential regulatory role of these DMRs/DhMRs in IBD-induced carcinogenesis of the colon.

Both DMRs and DhMRs were widely distributed across different genomic features (Figure 3A). Of these, 31.9% of DMRs and 26.7% of DhMRs were identified in the promoter region, 37% of DMRs and 40% of DhMRs were identified in the gene body, 25.5% of DMRs (28.9% of DhMRs) took place in the distal intergenic region, 0.9% of DMRs (1.4% of DhMRs) were found in 5′ UTR, 3.6% of DMRs (2.4% of DhMRs) were identified in 3′ UTR, and 1% of DMRs (0.6% of DhMRs) were located close to the downstream gene region (within 300 bp). Interestingly, IBD frequently led to increased cytosine methylation in gene promoter regions (Figure 3A). On the other hand, increased and decreased hydroxymethylation regions were distributed similarly across various genomic features (Figure 3A).

With the ability to separately measure cytosine methylation and hydroxymethylation, we were able to identify 104 CpG sites showing gain of both 5mC and 5hmC (quadrangle I in Figure 3B), 815 sites with de novo gain of 5hmC (quadrangle II in Figure 3B), 800 sites switching from 5mC to 5hmC (quadrangle III in Figure 3B), 3983 sites with a loss of 5mC (quadrangle IV in Figure 3B), 79 sites showing a loss of both 5mC and 5hmC (quadrangle V in Figure 3B), 590 sites characterized by the loss of 5hmC (quadrangle VI in Figure 3B), 838 sites switching from 5hmC to 5mC (quadrangle VII in Figure 3B), and 4730 sites with de novo gain of 5mC (quadrangle VIII in Figure 3B).

Genes characterized by de novo gain of 5hmC (*n* = 815, quadrangle II in Figure 3B) included *Creb3l1*. *Creb3l1* is involved in goblet cell differentiation and mucosa production in large intestine [53] as well as PI3K signaling in B lymphocytes, which plays an important role in B cell activation, differentiation, and survival [54]. Differentially modified CpG sites that underwent 5mC conversion to 5hmC (*n* = 800, quadrangle III in Figure 3B) showed an enrichment in thyroid cancer signaling pathway genes which include oncogenes such as *RET* (Table 1). *RET* fusions have been widely identified in CRC patients [55,56] and represent a subtype of CRC without *BRAF* and *RAS* mutations and high microsatellite instability [57]. CpG sites characterized by a loss of 5mC (*n* = 3983, quadrangle IV in Figure 3B) included cancer-related genes such as *Prkcb* and *Lin28b*. Elevated *Prkcb* expression induces enhanced cell proliferation and colon carcinogenesis in transgenic mice [58]. Overexpression of *Lin28* in a colon cancer mouse model leads to accelerated tumor formation and enhanced cell proliferation and invasiveness [59]. CpG sites characterized by a switch from 5hmC to 5mC (*n* = 838, quadrangle VII in Figure 3B) and de novo gain of 5mC (*n* = 4730, quadrangle VIII in Figure 3B) were both found preferentially in genes involved in basal cell carcinoma signaling, e.g., *Ctnnb1* and *Wnt6* (Table 1). *Ctnnb1* overexpression in an invasion front is a hallmark of colon cancer [60]. *Wnt6* is a gene in the Wnt/β-catenin pathway that can be activated through *Plagl2* and can promote cancer development in CRC [9]. Overall, our results reveal dynamic changes in cytosine methylation and hydroxymethylation patterns in the IBD mouse model. Importantly, these changes are enriched in genes involved in inflammatory response, cell proliferation, and colon carcinogenesis (Table 1).

### 2.4. Gene Expression Changes in IBD-Induced Hyperplasia/Dysplasia

To determine whether the observed changes in DNA methylation and hydroxymethylation (Figure 3) were accompanied by alterations in gene expression, RNA isolated from proximal colon tissues of infected and control mice was subjected to transcriptome profiling by RNA-seq. These analyses revealed significant gene expression changes in proximal colon tissues of IBD mice as compared to controls, with a total of 380 differentially expressed genes (222 upregulated and 158 downregulated, Figure 4A). Unsupervised hierarchical clustering based on 500 genes with the highest variance demonstrated a clear separation between treatment groups (control and infected mice, Figure 4A). This striking difference of gene expression profiling was also revealed by multidimensional scaling (Appendix A). 

IPA showed significant changes in the expression levels of genes associated with gastrointestinal disease (*Abcb1*, *Guca2a*, *Muc2*, *Rag2*, *Itga6*, and *Shh*), inflammatory disease (*Abcb1*, *Muc2*, and *Rag2*), and cancer (*Muc2*, *Guca2a*, and *Rag2*) (Figure 4C). Top upstream regulators identified by IPA included *Il10ra*, *Pla2g4a*, *Klf4*, and *Ncoa3*. IL10RA and IL10RB together form a heterotetramer receptor complex of IL10, which mediates the downstream activation of STAT3 and its anti-inflammatory response [61,62]. The network surrounding *Il10ra* shows that many of them are cancer related, e.g., *Reg3g*, *Aoc1*, *Mep1a*, *Irf7*, *Gas6*, *B3gnt7*, *Tap1*, *Tgm2*, and *Nos2* (Figure 4B).

*Pla2g4a* gene encodes cytosolic phospholipase A2-α, which catalyzes the hydrolysis of membrane phospholipid, yielding arachidonic acid. Reduced expression of *Pla2g4a* is common in gastric cancer and is significantly associated with tumor size and grade [63]. KLF4 inhibits the G1/S transition of the cell cycle and has been identified as a potential tumor suppressor gene in colorectal cancer [64]. NCO3 is a transcriptional coactivator that promotes CRC cell proliferation and invasiveness, partially through enhanced notch signaling [65].

To validate our RNA-seq results, RT-qPCR was conducted for a subset of genes including *Tet1-3*. In agreement with RNA-seq results, *Tet1-3* transcript amounts were decreased in the IBD group compared to controls, although the differences were not statistically significant (Appendix A). Decreased expression of TET proteins may account for the global decrease in 5hmC abundance (Figure 1D) and the overall dysregulation of the methylome and the hydroxymethylome (Figure 2).

### 2.5. Integrated Analysis Identified Differentially Expressed Genes Aberrantly Regulated by DNA Methylation

In order to identify epigenetic drivers of IBD induced hyperplasia/dysplasia in our animal model, further data analysis was conducted focusing on genes that exhibited changes in cytosine methylation/hydroxymethytlation and that were also differentially expressed (Figure 5A,B). These analyses identified a number of cancer-related genes. For instance, *Duox2* was upregulated in IBD induced hyperplasia/dysplasia, which was accompanied by *Duox2* promoter hypomethylation (Figure 5A). DUOX2, when combined with its maturation factor DUOXA2, will produce potentially mutagenic hydrogen peroxide (H_2_O_2_) [66]. Hydrogen peroxide is a reactive oxygen species capable of inducing oxidative DNA damage [66] and is known to contribute to chemoresistance-induced epithelial-to-mesenchymal transition in colon cancer cells [67]. We also observed decreased promoter hydroxymethylation and upregulation of *Hk2*, which catalyzes the phosphorylation of glucose to yield glucose-6-phosphate in glycolysis and is required for tumor initiation [68] (Figure 5B). In addition, *Aldh1l1*, which is known to be downregulated in cancer through promoter methylation [69,70], is upregulated in our IBD model and exhibits two regions with reduced cytosine hydroxymethylation. Our analyses also identified a number of genes, the function of which is not yet clear in IBD-induced colon cancer (Figure 5 and Appendix A).

## 3. Discussion

Multiple studies have revealed dysregulated DNA methylome in IBD [37,71,72], however, limited attention had been paid to changes in DNA hydroxymethylome. As discussed above, traditional bisulfite sequencing methodologies are unable to distinguish between 5hmC and 5mC, leading to confounded profiling of the two epigenetic marks [45]. This limits our understanding of epigenetic changes in IBD. 5hmC is an important demethylation intermediate [27,28,29,30] and an epigenetic mark in its own right [31,73].

To our knowledge, our study was the first to characterize IBD-associated changes in DNA methylome and hydroxymethylome at single base resolution. *Rag2^−/−^/Il10^−/−^* mice infected with *H. hepaticus* are an established animal model of chronic inflammation in the lower bowel and IBD [10,21]. A recently developed oxRRBS/RRBS methodology [46] was employed to separately map DNA methylation and hydroxymethylation changes in the proximal colon of infected mice. Our results revealed a number of differentially methylated and hydroxymethylated genomic regions (DMRs and also DhMRs), many of which are associated with gastrointestinal diseases and inflammatory response (Figure 2D,E). RNA-Seq analyses revealed profound changes in the patterns of gene expression (Figure 3A). Overall, our results suggest that both DNA epigenetic marks are affected in IBD, likely contributing to disease pathology.

Our methylation results are consistent with previously identified DMRs in human IBD patients, such as increased methylation in the promoter of *GATA3* [74], *THBS1*, and *CDH13* [75]. It should be noted that the observed methylation changes were nonuniform throughout the gene. For instance, the *Gata3* promoter was hypermethylated in IBD while its third intron was hypomethylated (Appendix A). In addition, inflammation in the proximal colon led to increased hydroxymethylation of the promoter region of *Thbs1* and increased hydroxymethylation of the third intron of *Cdh13* (Appendix A). These results reveal a complex epigenetic regulatory network at the target genes, not only by 5mC but also potentially by 5hmC.

RNA-seq analyses revealed significant changes in gene expression profiles in the proximal colon tissues of *Rag2*^−/−^/*Il10*^−/−^ mice with IBD. This is not surprising considering the genetic background of these mice. *Il10* is required for regulatory lymphocytes to suppress the innate inflammatory response and the associated carcinogenesis [21]. In addition, *Il10* downregulates *Tnf*-*alpha*, which could trigger colonic inflammation and carcinogenesis [10]. Therefore, this new model (*Rag2^−/−^/Il10^−/−^* double knockout) is even more likely to develop inflammation-associated cancer compared to *Rag2^−/−^* mice used previously [10]. IPA revealed that genes exhibiting expression changes in response to inflammation in proximal colon were associated with gastrointestinal disease, cancer, inflammatory response, cellular growth, and proliferation (Figure 4C).

The top upstream regulator in RNA-seq identified by IPA includes interleukin 10 receptor, alpha (*Il10ra*). In the network surrounding *Il10ra* (Figure 4B), many of the downstream differentially expressed genes are cancer related. For instance, RNA-seq revealed significantly higher expression levels of *Nos2* (*iNOS*) in IBD (Figure 4B), which is in agreement with previous report [12]. *Nos2* (nitric oxide synthetase 2) generates nitric oxide, which can induce nitrosative and oxidative stress and can contribute to inflammation-induced carcinogenesis [76,77,78]. We also detected an increase in expression of serum amyloid A 3 (Saa3), a family member of Saa protein family (Appendix A). This protein is essential for the protumorigenic property of the cancer-associated fibroblast cells in pancreatic cancer [79] and has been reported to be the most increased protein from a previous proteomic study using serum from patients with ulcerative colitis [11].

Side by side comparison of our RNA-seq and RRBS-oxRRBS results revealed a modest overlap between differentially methylated/hydroxymethylated regions and differentially expressed genes (Figure 5A,B). It should be noted that changes in gene expression can be induced by epigenetic mechanisms distinct from DNA methylation/hydroxymethylation, e.g., histone modifications [80] and expression of noncoding RNA [81]. Cytosine methylation within CpG islands of gene promoter regions is traditionally considered a repressive mark, while gene body hydroxymethylation has been linked with gene activation [82,83,84]. However, recent studies reveal a more complex picture [85,86]. In our dataset, the relationship between methylation/hydroxymethylation and gene expression is highly gene-dependent (Figure 5A,B). When all DMR/DhMR are considered, data analysis reveals no direct correlation between gene expression and gene methylation/hydroxymethylation status (Figure 5A,B). However, this does not rule out the local effects of cytosine methylation/hydroxymethylation status on expression levels of specific genes; it simply suggests that such effects are gene-dependent and cannot be generalized across the entire genome.

From the list of genes that show both differential methylation/hydroxymethylation and differential expression, we identified a number of genes that are known to be differentially expressed in human colorectal cancer. For instance, Duox2 is known to promote the progression of CRC [87], Tgm2 is overexpressed in human CRC tissue and is associated with increased cell proliferation [88], Cdhr5 is epigenetically downregulated in colorectal tumors [89], Hk2 is overexpressed in human CRC tissues and could potentially be a target for clinical treatment of CRC [90], and low expression of Gcnt3 is a high risk factor for relapse [91]. In the murine IBD model with significant dysplasia and hyperplasia prior to solid tumor formation, these results suggest an early effect of DNA epigenetic marks on triggering altered expression of cancer-related genes. Our recent study involving a murine model of inflammation-induced lung cancer analogously revealed early epigenetic changes within tumor suppressor genes Rassf1, Cdh13, and Dapk1 [52]. Future functional studies will establish whether such epigenetic dysregulation directly contributes to initiation and progression of cancer.

The main limitations of our study include its relatively small sample size (*n* = 3 per group) and the use of whole tissue samples rather than specific cell types. As a result, the reported RNA-seq, DNA methylation, and hydroxymethylation data refer to the average readouts from all cells within the proximal colon tissue. There is a possibility that the complicated cell composition in colon tissue could mask epigenetic changes in specific cell types. In addition, other layers of epigenetic regulation such as histone modifications, chromatin structure, and expression of noncoding RNA were not characterized here. Finally, functional validation of the candidate genes that exhibit changes in epigenetic marks and expression levels remain to be done. Such experiments should be conducted in the future in order to obtain a more comprehensive understanding of the epigenetic drivers in colon cancer.

Upon considering IBD-induced changes in methylome, hydroxymethylome, and transcriptome, the following mechanistic model for the role of epigenetics in IBD-induced CRC can be proposed (Figure 5C). Inflammation in proximal colon induces site-specific changes in cytosine methylation and hydroxymethylation. Some of these DMR/DhMR can be integrated with differential gene expression at RNA level (Figure 5A,B). These genes play an important role in cell turnover (e.g., *Tgm2* [88], *Hk2* [90], and *Cdhr5* [92]) and mutagenesis (*Duox2*) [66], potentially contributing to CRC initiation (Figure 5C). Taken together, our results identify early inflammation-related changes in DNA methylation and hydroxymethylation in the colon and point to a number of epigenetically controlled candidate genes that deserve further investigation for their role in in colon cancer etiology and as potential drug targets for future anticancer therapies.

## 4. Materials and Methods

### 4.1. IBD Mouse Model

All experiments were performed in accordance with protocols approved by the Massachusetts Institute of Technology Committee on Animal Care and with the National Institutes of Health Guide for the Care and Use of Laboratory Animals (protocol 0909-093-12, 14 September 2009). 129S6/SvEvTac*-Rag2^tm1Fw^Il10^−/−^* (*Rag2^−/−^/Il10^−/−^*) mice were used and housed in an Association for Assessment and Accreditation of Laboratory Animal Care-accredited specific pathogen-free barrier facility. Mice were maintained in polycarbonate microisolator caging (Allentown Caging Equipment Inc., Allentown, NJ, USA) and provided pelleted diet (ProLab 3000, Purina Mills, St. Louis, MO, USA) and filtered water ad libitum. Mice were infected with *H. hepaticus* (strain 3B1, ATCC 51449) as described previously [10]. Mice aged 6–7 weeks received 0.2 mL of 2 × 10^8^ fresh inoculums (infected group: 7 males and 7 females) or sterile media (control group: 7 males and 7 females) by gavage every other day for a total of three doses. During the course of infection, the overall health of infected mice declined rapidly and select cages were maintained on the pelleted diet soaked in water to prevent dehydration. Mice were infected with *H. hepaticus* (strain 3B1, ATCC51449) grown and confirmed as a pure culture as described previously [21,49]. Mice were sacrificed 20 weeks post infection, and colon tissues were collected by standard necropsy procedures. For histopathological evaluation, the entire large intestine including the ileocecocolic junction and colon were harvested from each mouse.

### 4.2. Histopathology Analysis

During postmortem examination, sections of tissue from different regions of the cecum (greater curvature and ileocecal junction) and colon (proximal, transverse, and distal) were collected and immediately fixed in 10% neutral-buffered formalin. Tissue sections were embedded in paraffin, processed into 4-µm sections, and stained with hematoxylin and eosin. All slides were examined by a board-certified veterinary pathologist (N.M.A.P.) who was blinded to sample identity [93,94].

At each of the separate locations within the cecum and colon, lesions of inflammation, edema, epithelial defects, crypt atrophy, hyperplasia, and dysplasia were scored. Lesion severity score was assigned using an ascending scale from 0 to 4: 0 (absent), 1 (mild), 2 (moderate), 3 (marked), and 4 (severe). The histological activity index was then calculated at each location as the sum of scores for inflammation, edema, epithelial defects, crypt atrophy, hyperplasia, and dysplasia on a scale of 0–4 (maximum score 24) [94].

### 4.3. HPLC-ESI^+^-MS/MS Quantitation of Total 5mC and 5hmC

Following extraction, genomic DNA (2–4 µg) was subjected to hydrolysis with PDE I (3.6 U), PDE II (3.2 U), DNase I (50U), and alkaline phosphatase (10 U) in 10 mM Tris HCl/15 mM MgCl_2_ buffer (pH 7) at 37 °C overnight. The hydrolysates were spiked with ^13^C_10_^15^N_2_-5-methyl-2-deoxycytidine (1 pmol) and 5-hydroxymethyl-d_2_-deoxycytidine-6-d_1_ (1 pmol) (internal standards for mass spectrometry) [95] and filtered through Nanosep 10K Omega filters (Pall Corporation, Port Washington, NY, USA) to remove proteins.

The hydrolysates were dried and separated by offline HPLC to enrich for 5mC and 5hmC as follows. An Atlantis T3 column (4.6 × 150 mm, 3 µm, Waters, Milford, MA, USA) was eluted at a flow rate of 0.9 mL/min with a gradient of 5 mM ammonium formate buffer, pH 4.0 (A), and methanol (B). Solvent composition was changed linearly from 3 to 30% B over 15 min, increased to 80% over the next 3 min, maintained at 80% B for the next 2 min, and brought back to 3% B. The column was equilibrated at 3% B for 7 min. dC was quantified by HPLC-UV using calibration curves obtained by analyzing authentic dC standards. HPLC fractions corresponding to 5mC and 5hmC (7–8.6 min for 5hmC and 9–10.5 min for 5mC) were combined, dried, and analyzed by isotope dilution HPLC-ESI-MS/MS.

Quantitation of 5mC and 5hmC was performed using a Dionex Ultimate 3000 UHPLC (Thermo Fisher, Waltham,MA, USA) interfaced with a Thermo TSQ Quantiva mass spectrometer (Thermo Fisher, Waltham, MA, USA). Chromatographic separation was achieved on a Zorbax SB-C18 column (0.5 × 150 mm, 3 µm, Agilent, Santa Clara, CA, USA) eluted at a flow rate of 10 μL/min with a gradient of 2 mM ammonium formate (A) and acetonitrile (B). Solvent composition was linearly changed from 3 to 5% B over 6 min, then linearly increased to 43.5% B over the next 7 min, and maintained at 43.5% B for 1 min. The solvent composition was then returned to initial conditions (3% B) and re-equilibrated for 7 min. Under these conditions, 5mC and ^13^C_10,_^15^N_2_-5mC eluted at 4.1 min, while 5hmC and its internal standard (d_3_-hmC) eluted at 3.5 min. Quantitation was achieved by monitoring the transitions *m*/*z* 258.1 [M + H^+^] → 141.1 [M—deoxyribose + H^+^] for 5hmC, *m*/*z* 261.2 [M + H^+^] → *m*/*z* 145.1 [M—deoxyribose + H^+^] for D^3^-5hmC, *m*/*z* 242.1 [M + H^+^] → 126.1 [M + H^+^] for 5mC, and *m*/*z* 254.2 [M + H^+^] → 133.1 [M + H^+^] for ^13^C_10,_^15^N_2_-5mC. Mass spectrometry parameters were optimized while infusing authentic standards of 5mC and 5hmC. Typical MS settings were a spray voltage of 3000 V, a sheath gas of 15 units, the declustering voltage at 5 V, the RF lens at 40 V, and the ion transfer tube temperature at 350 °C. The full-width at half-maximum was maintained at 0.7 for both Q1 and Q3. MS/MS fragmentation was induced using a collision gas of 1.0 mTorr and a collision energy of 10.3 V.

### 4.4. RNA Sequencing

Genomic DNA and total RNA were extracted from approximately 10 mg of proximal colon tissue using the AllPrep DNA/RNA mini kit (QIAGEN, Hilden, Germany). RNA-seq libraries were prepared by Clontech Smarter Stranded Total RNA-Seq Kit (Takara Bio Inc., Kusatsu, Japan) according to the manufacturer’s instruction. RNA libraries were sequenced at the University of Minnesota Genomics Center on an Illumina HiSeq 2500 instrument, with an average coverage of 35 million, 50 bp paired reads. The raw sequencing reads and processed file have been uploaded to the GEO database (GSE163036 under GSE163038).

### 4.5. Bioinformatics Analysis of RNA-Seq Data

RNAseq reads were cleaned of adapter contamination, and low-quality bases were cleaned with Trimmomatic 0.33 [96]. Cleaned reads were aligned to the mouse mm10 genome with HISAT2 2.1.0 [97] using an index that was built with known splice sites and known single nucleotide polymorphisms. Only reads in complete pairs were aligned to the genome.

Fragments that aligned to annotated genes were counted with the “featureCounts” program in the subread package [98]. Raw counts for each gene and each sample were transformed into counts-per-million values with the edgeR package [99]. Differentially expressed genes were identified with a quasi-likelihood test in edgeR [100], with a 0.05 false discovery rate correction applied.

### 4.6. RRBS and oxRRBS

Genomic DNA (100 ng) was subjected to RRBS [101] and oxRRBS [46] to map 5mC and 5hmC, respectively, across the genome. RRBS and oxRRBS libraries were prepared by Ovation^®^ RRBS Methyl-Seq with the TrueMethyl^®^ oxBS kit (Tecan, Redwood City, CA, USA) according to the manufacturer’s instructions. The resulting libraries were sequenced at the UMGC on an Illumina Nextseq 550 instrument, with an average coverage of 33 million, 75 bp paired reads. The raw sequencing reads and processed files have been uploaded to GEO database (GSE163037 under GSE163038).

### 4.7. RRBS Bioinformatics Analysis

Reduced representation bisulfite sequencing (RRBS) reads and oxRRBS reads were trimmed of low-quality bases and adapter contamination with TrimGalore! 0.4.4_dev (http://www.bioinformatics. babraham.ac.uk/projects/trim_galore/) in paired-end mode using Cutadapt version 1.8.1. (https://journal.embnet.org/index.php/embnetjournal/article/view/200). Any read pairs where one mate failed quality control were discarded. Cleaned reads were aligned to the mouse mm10 genome with Bismark version 0.19.0 [102]. Resulting alignments were cleaned of PCR duplicates with NuDup (https://github.com/nugentechnologies/nudup) and reads with mapping quality of less than 20 were removed.

Analysis of methylated and hydroxymethylated regions followed the Methpipe analysis pipeline [103]. Bisulfite conversion rates were estimated with the “bsrate” program in Methpipe. Genome-wide cytosines were filtered to represent only symmetrically methylated CpG sites. CpG sites that were not covered by at least 10 reads in each sample were discarded. CpG sites were further filtered to exclude sites that occurred within “blacklisted” regions of the mm10 genome assembly (ENCODE file ID ENCFF547MET). Differentially methylated (DMR) and hydroxymethylated (DhMR) regions were identified by merging consecutive CpGs that crossed the threshold for statistical significance at a false discovery rate of 0.05, and filtering regions with at least three significant CpG sites. We then used Chipseeker [104] to annotate the identified DMR and DhMR.

### 4.8. RT-qPCR

The first-strand complementary DNA was synthesized by using SuperScript IV VILO Master Mix (Invitrogen, Waltham, MA, USA). RT-qPCR was performed by StepOnePlus Real-Time PCR System (Applied Biosystems, Foster City, CA, USA) using SYBR Select Master Mix (Applied Biosystems, Foster City, CA, USA) and gene-specific primers (Table 2). The 2ˆ^−∆*C*T^ value was used to show the relative gene expression that was normalized to an internal control gene, β-actin.

## Figures and Tables

**Figure 1 ijms-22-00364-f001:**
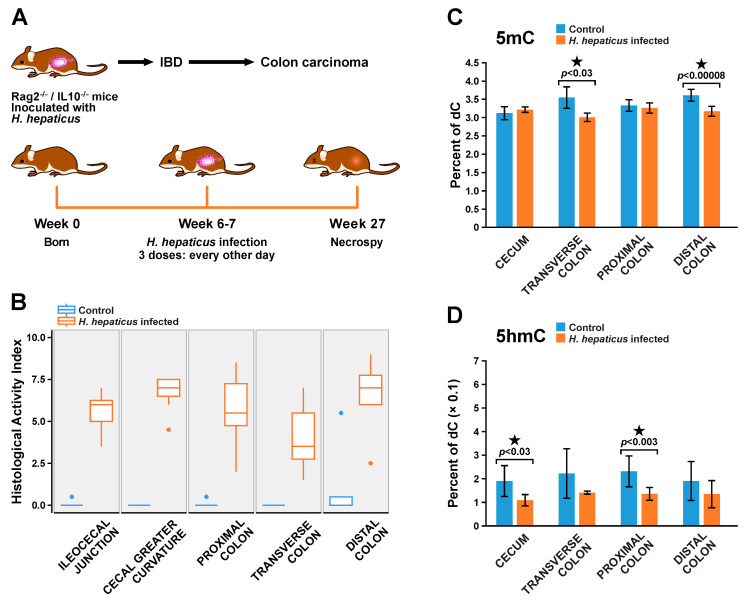
(**A**) Animal study design: *Rag**2*^−/−^/*Il**10*^−/−^ mice were inoculated with either *H. hepaticus* or sterile media (control) at 6–7 weeks of age and were sacrificed 20 weeks post infection. (**B**) Histopathology results for colon tissues of *Rag2^−/−^/Il10^−/−^* male mice inoculated with *H. hepaticus* (orange) and control mice treated with saline only (blue): the histological activity index was calculated by summing individual scores of inflammation, edema, epithelial defects, crypt atrophy, hyperplasia, and dysplasia for different parts of gastrointestinal tract (*n* = 7). (**C**,**D**) Global 5-methylcytosine (5mC) (**C**) and 5-hydroxymethyl-cytosine (5hmC) (**D**) changes in different parts of the gastrointestinal tract of *Rag**2*^−/−^/*Il**10*^−/−^ male mice with or without *H. hepaticus* infection (*n* = 4, for each group in cecum, *n* = 3, for each group in transverse colon, and *n* = 7 for each group in proximal and distal colon).

**Figure 2 ijms-22-00364-f002:**
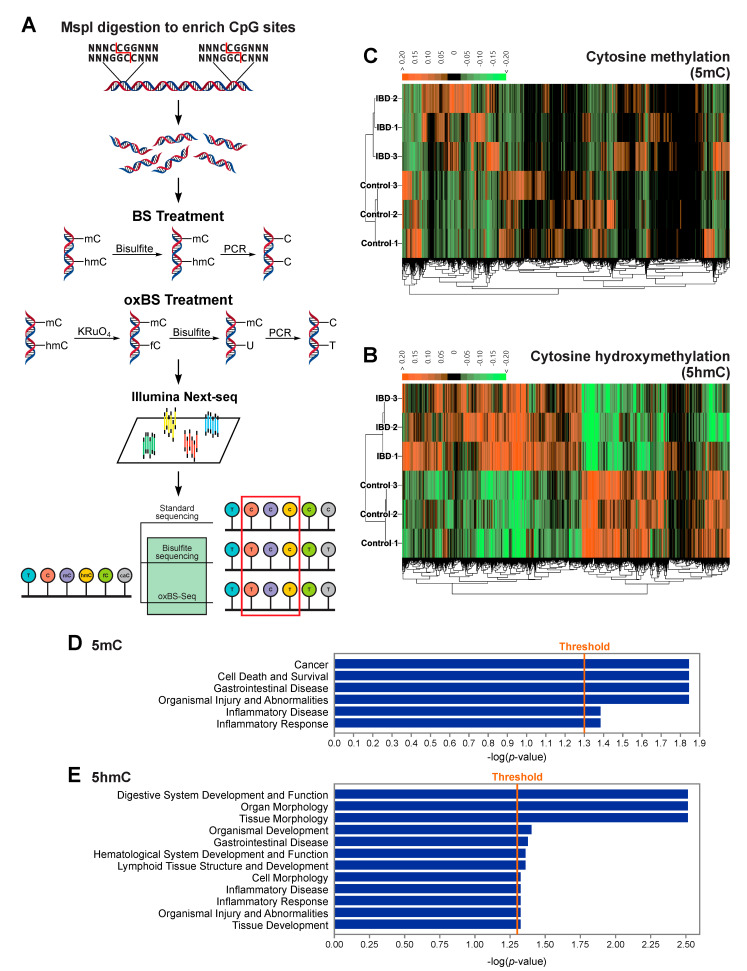
Differentially methylated region (DMR) and differentially hydroxymethylated region (DhMR) containing genes are enriched in genes associated with carcinogenesis and gastrointestinal disease. (**A**) A schematic illustration of the reduced representation bisulfite sequencing (RRBS) and oxidative RRBS (oxRRBS) techniques: 5mC and 5hmC fractions are calculated through a comparison of the RRBS and oxRRBS datasets. (**B**,**C**) Heatmaps showing 1606 DMRs and 3011 DhMRs that have been identified through RRBS combined with oxRRBS. (**D**,**E**) Functions and diseases predicted to be affected by genes containing DMRs (**D**) and DhMRs (**E**), using Fisher’s exact test at a significance threshold of *p* = 0.05, via Ingenuity Pathway Analysis (IPA).

**Figure 3 ijms-22-00364-f003:**
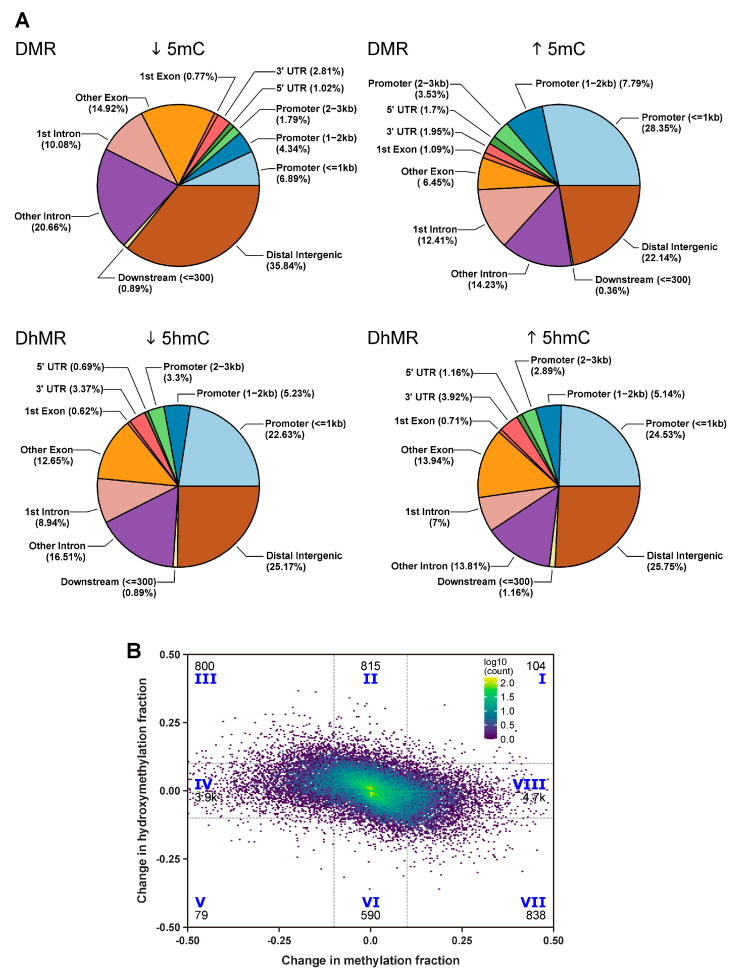
Dynamic methylation and hydroxymethylation changes at single-nucleoside resolution upon *H. hepaticus*-induced inflammation in *Rag2^−/−^/Il10^−/−^* mice: (**A**) pie charts showing the genomic feature distribution of 784 hypo DMRs, 822 hyper DMRs, 1454 hypo DhMRs, and 1557 hyper DhMRs and (**B**) dynamic change of 5mC and 5hmC at each CpG sites within DMR/DhMR. We considered 0.1 as a cutoff of significant change in the 5mC/5hmC fraction. Numbers represent counts of sites.

**Figure 4 ijms-22-00364-f004:**
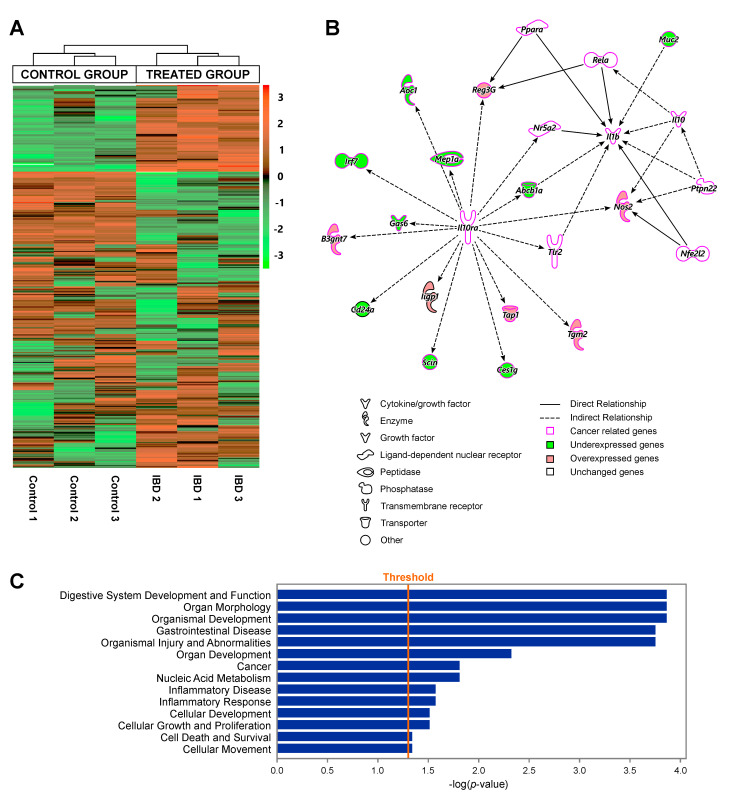
Transcriptome profiling of proximal colon in inflammatory bowel disease (IBD) and control mice: (**A**) unsupervised hierarchical clustering based on the 500 genes with the highest variances, showing a clear separation between control and IBD groups; (**B**) the network surrounding the top upstream regulator *Il10ra* as identified by IPA; and (**C**) diseases and functions predicted to be affected by differentially expressed genes, using Fisher’s exact test at a significance threshold of *p* = 0.05, via IPA.

**Figure 5 ijms-22-00364-f005:**
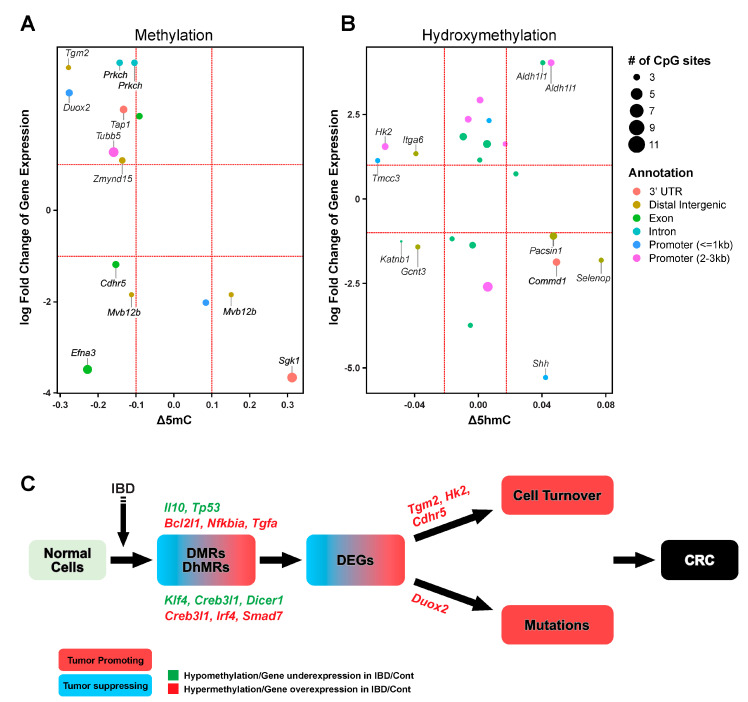
Correlation between IBD-induced changes in the transcriptome, DNA methylome, and hydroxymethylome: (**A**,**B**) the scatterplots show genes that are both differentially expressed and that contain DMRs (**A**) or DhMRs (**B**). Colors indicate different genomic features, and the dot size indicates the number of CpGs in each region. (**C**) A model summarizing the contribution of DNA epigenetic marks in the initiation of IBD-induced colorectal cancer (CRC)**.**

**Table 1 ijms-22-00364-t001:** Top canonical pathways from IPA of CpG sites showing differential methylation and hydroxymethylation upon *H. hepaticus*-induced inflammation in *Rag2^−/−^/Il10^−/−^* mice.

**Gain of Both 5mC and 5hmC (Δ5mC > 0.1 and Δ5hmC > 0.1)**	
**Name**	***p*-value**
Agrin Interactions at Neuromuscular Junction	1.28 × 10^−2^
Axonal Guidance Signaling	1.47 × 10^−2^
Agranulocyte Adhesion and Diapedesis	3.77 × 10^−2^
Maturity Onset Diabetes of Young (MODY) Signaling	4.53 × 10^−2^
RhoGDI Signaling	4.67 × 10^−2^
**De novo gain of 5hmC (−0.1 < Δ5mC < 0.1 and Δ5hmC > 0.1)**	
**Name**	***p*-value**
Corticotropin Releasing Hormone Signaling	2.14 × 10^−3^
Axonal Guidance Signaling	2.38 × 10^−3^
PI3K Signaling in B Lymphocytes	2.92 × 10^−3^
Sperm Motility	5.15 × 10^−3^
Hepatic Fibrosis Signaling Pathway	6.28 × 10^−3^
**Switching from 5mC to 5hmC (Δ5mC < −0.1 and Δ5hmC > 0.1)**	
**Name**	***p*-value**
Cellular Effects of Sildenafil (Viagra)	3.93 × 10^−4^
Sperm Motility	7.18 × 10^−4^
Aryl Hydrocarbon Receptor Signaling	7.98 × 10^−4^
Thyroid Cancer Signaling	2.82 × 10^−3^
Glutathione Redox Reactions II	4.01 × 10^−3^
**Loss of 5mC (Δ5mC < −0.1 and −0.1 < Δ5hmC < 0.1)**	
**Name**	***p*-value**
Thyroid Cancer Signaling	1.11 × 10^−4^
Synaptogenesis Signaling Pathway	3.55 × 10^−4^
Aryl Hydrocarbon Receptor Signaling	7.81 × 10^−4^
Corticotropin Releasing Hormone Signaling	1.45 × 10^−3^
Gαs Signaling	2.13 × 10^−3^
**Loss of both 5mC and 5hmC (Δ5mC < −0.1 and Δ5hmC < −0.1)**	
**Name**	***p*-value**
Amyotrophic Lateral Sclerosis Signaling	1.21 × 10^−2^
Endocannabinoid Neuronal Synapse Pathway	1.26 × 10^−2^
Aryl Hydrocarbon Receptor Signaling	3.21 × 10^−2^
Endocannabinoid Cancer Inhibition Pathway	3.28 × 10^−2^
Breast Cancer Regulation by Stathmin1	3.69 × 10^−2^
**Loss of 5hmC (−0.1 <Δ5mC < 0.1 and Δ5hmC <−0.1)**	
**Name**	***p*-value**
Netrin Signaling	2.05 × 10^−4^
Synaptogenesis Signaling Pathway	9.64 × 10^−4^
Apelin Cardiac Fibroblast Signaling Pathway	2.89 × 10^−3^
Dopamine-DARPP32 Feedback in cAMP Signaling	3.32 × 10^−3^
Apelin Muscle Signaling Pathway	3.62 × 10^−3^
**Switching from 5hmC to 5mC (Δ5mC > 0.1 and Δ5hmC < −0.1)**	
**Name**	***p*-value**
HOTAIR Regulatory Pathway	1.28 × 10^−4^
Wnt/β-catenin Signaling	1.73 × 10^−4^
Basal Cell Carcinoma Signaling	2.26 × 10^−4^
Endocannabinoid Neuronal Synapse Pathway	5.67 × 10^−4^
Transcriptional Regulatory Network in Embryonic Stem Cells	1.24 × 10^−3^
**De novo gain of 5mC (Δ5mC > 0.1 and −0.1 < Δ5hmC< 0.1)**	
**Name**	***p*-value**
Axonal Guidance Signaling	7.42 × 10^−7^
Wnt/β-catenin Signaling	4.29 × 10^−5^
Role of NANOG in Mammalian Embryonic Stem Cell Pluripotency	1.28 × 10^−4^
Basal Cell Carcinoma Signaling	3.33 × 10^−4^
Osteoarthritis Pathway	4.08 × 10^−4^

**Table 2 ijms-22-00364-t002:** Primers for RT-qPCR.

Gene Name	Forward Primer (5′-3′)	Reverse Primer (5′-3′)
*β-actin*	ACTCTTCCAGCCTTCCTTCC	GTACTTGCGCTCAGGAGGAG
*Tet1*	AGATGGCTCCAGTTGCTTATC	CTTCCGTTGTGCATGTTGTG
*Tet2*	GTCCTGATGTGGCAGCTATT	TCCTCACTCGATCTCCGATATAC
*Tet3*	GAGTTCCCTACCTGCGATTG	TCCATGAGTTCCCGGATAGA

## Data Availability

The data presented in this study are openly available in GEO database, accession number [GSE163038].

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
