# Peer review of "Multi-Omics Characterization of Inflammatory Bowel Disease-Induced Hyperplasia/Dysplasia in the Rag2−/−/Il10−/− Mouse Model"

_ijms, 2020, doi:10.3390/ijms22010364_

Round 1
Reviewer 1 Report
- Why in a study focused on IBD you used a hepatitis score too? (“The liver tissues were scored according to previously defined criteria, also using an ascending scale from 0 to 4. Sections from each of the different liver lobes were individually scored for lobular, portal, and interface hepatitis. A hepatitis index (HI) was then calculated for each animal, by combining the three hepatitis scores with the number of lobes (out of a total of 4) that contained 5 or more inflammatory foci. Hepatitis was defined by a HI equal to or greater than 4”)
- The result section can be simplified to be more readable
- Figure 5 is cut
- How many cases and controls did you include? How did you calculate the statistical power of the study?
- Add the criticisms to your study
Author Response
- Why in a study focused on IBD you used a hepatitis score too? (“The liver tissues were scored according to previously defined criteria, also using an ascending scale from 0 to 4. Sections from each of the different liver lobes were individually scored for lobular, portal, and interface hepatitis. A hepatitis index (HI) was then calculated for each animal, by combining the three hepatitis scores with the number of lobes (out of a total of 4) that contained 5 or more inflammatory foci. Hepatitis was defined by a HI equal to or greater than 4”)
Response: We have removed this description from the Methods section.
- The result section can be simplified to be more readable
Response: We have simplified and rephrased the description of our results to be more readable. The changes are all over the result section but can be checked under reviewer mode with showing all markup.
- Figure 5 is cut
Response: We have adjusted the size and the position of Figure 5.
- How many cases and controls did you include?
Response: We have included 3 control mice and 3 H.hepaticus infected mice for both RNA-seq and RRBS/oxRRBS experiment. As described below, the number of replicates for RRBS/oxRRBS was limited by the capacity and the cost of commercial RRBS/oxRRBS kits.
- How did you calculate the statistical power of the study?
Response: In the planning stages of this study, we used the RNA-seq calculator(https://cqs-vumc.shinyapps.io/rnaseqsamplesizeweb/)1 to calculate the sample size needed to achieve 80% power. Based on our prior data, we have set that the minimum average read counts among the prognostic genes in the control group is 167, the maximum dispersion is 0.2, and the ratio of the geometric mean of normalization factors is 1, assuming that the total number of genes for testing is 12000 and the top 300 genes are prognostic. If the desired minimum fold change is 2, we would need to study 15 subjects in each group to be able to reject the null hypothesis that the population means of the two groups are equal with probability (power) 0.8 using exact test. The FDR associated with this test of this null hypothesis is 0.05.
To determine the statistical power of the RRBS RRBS/oxRRBS study, we employed an RRBS estimator2. We found that with 30 samples (15 per each group), the power to detect true positives would be just below 6%. The sample size would need to be even much larger than that in order to reach 80% power, which would be prohibitively expensive and not practical. Indeed, statistical power calculation did not have too much practical meaning in our case due to the prohibitive cost of conducting 30 RRBS/oxRRBS experiments, which is far beyond the limit of feasibility.
When comparing our study design with published work, we realized that nearly every published study that used animal models and NGS technology is "underpowered" because they are so expensive and laborious to perform. For example, Brant, et al. examined the influence of the Prader-Willi syndrome imprinting center on the DNA methylation landscape in the mouse brain3, Mellen et al. reported 5-hydroxymethylcytosine accumulation in postmitotic neurons results in functional demethylation of expressed genes 4, while Haldy et al. examined the epigenetic drivers of hepatocellular carcinoma5. All of these studies did not report statistical power calculation and employed small sample sizes. Instead, they have focused on the strongest observed differences rather than applying a strict significance cutoff.
With N = 3, we observed significant changes in cytosine methylation/hydroxymethylation and gene expression (Figures 2-5). Although study of this size is likely to include some false positives, further validation and functional studies will follow to confirm or reject these initial findings. This information has been added to the Discussion section of the revised manuscript.
- Add the criticisms to your study
Response: We have added this statement to the Discussion section (Page 14, line 530-539):
The main limitations of our study include its small sample size and the use of whole tissue samples rather than specific cell types. As a result, the reported RNA-seq, DNA methylation, and hydroxymethylation data refer to average readouts from all cells within the colon tissue. There is a possibility that the complicated cell composition in colon tissue could mask epigenetic changes in specific cell types. In addition, other layers of epigenetic regulation such as histone modifications, chromatin structure, and expression of noncoding RNA were not characterized here. Finally, functional validation of the candidate genes that exhibit changes in epigenetic marks and expression levels should be conducted in the future. Such experiments should be conducted in the future in order to obtain a more comprehensive understanding of the epigenetic drivers in colon cancer.
- Guo, Y.; Zhao, S.; Li, C. I.; Sheng, Q.; Shyr, Y., RNAseqPS: A Web Tool for Estimating Sample Size and Power for RNAseq Experiment. Cancer Inform 2014, 13 (Suppl 6), 1-5.
- Lea, A. J.; Vilgalys, T. P.; Durst, P. A. P.; Tung, J., Maximizing ecological and evolutionary insight in bisulfite sequencing data sets. Nat Ecol Evol 2017, 1 (8), 1074-1083.
- Brant, J. O.; Riva, A.; Resnick, J. L.; Yang, T. P., Influence of the Prader-Willi syndrome imprinting center on the DNA methylation landscape in the mouse brain. Epigenetics 2014, 9 (11), 1540-56.
- Mellén, M.; Ayata, P.; Heintz, N., 5-hydroxymethylcytosine accumulation in postmitotic neurons results in functional demethylation of expressed genes. Proc Natl Acad Sci U S A 2017, 114 (37), E7812-E7821.
- Hlady, R. A.; Sathyanarayan, A.; Thompson, J. J.; Zhou, D.; Wu, Q.; Pham, K.; Lee, J. H.; Liu, C.; Robertson, K. D., Integrating the Epigenome to Identify Drivers of Hepatocellular Carcinoma. Hepatology 2019, 69 (2), 639-652.
Reviewer 2 Report
In the present basic science article Han et al found, in a transgenic mouse model of inflammatory bowel disease (IBD), that a different pattern of methylation of some genes (Duox2, Tgm2…) may be found in the different stages of colonic inflammation and in relation to dysplastic changes.
The paper is well written and the study was well planned. I have only two objections:
1) It is unclear whether all mice had dysplastic areas and, therefore, these results refer to inflammation, dysplasia or both.
2) There are too many abbreviations that make the paper hard to read. In order to improve this problem, they should be all listed in the dedicated paragraph at page 17.
Author Response
In the present basic science article Han et al found, in a transgenic mouse model of inflammatory bowel disease (IBD), that a different pattern of methylation of some genes (Duox2, Tgm2…) may be found in the different stages of colonic inflammation and in relation to dysplastic changes.
The paper is well written and the study was well planned. I have only two objections:
1) It is unclear whether all mice had dysplastic areas and, therefore, these results refer to inflammation, dysplasia or both.
Response: All the H.hepaticus infected mice had hyperplasia and dysplasia as shown in Table S1 in the supplementary file, while the tissues of control mice are normal, so our result refers to inflammation induced hyperplasia/dysplasia. Inflammation is the cause and hyperplasia/dysplasia are the phenotype. In order to reduce confusion, we changed the title of result section 2.2 and 2.3 and 2.5.
2) There are too many abbreviations that make the paper hard to read. In order to improve this problem, they should be all listed in the dedicated paragraph at page 17.
Response: We have listed additional abbreviations at page 17 in the revised version and we deleted abbreviations appearing no more than 3 times.
RRBS/oxRRBS: Reduced representation bisulfite sequencing/ oxidative RRBS
H.hepaticus: Helicobacter hepaticus
DNMT: DNA methyltransferase
TET: Ten-eleven translocation dioxygenases
5fC: 5-formylcytosine
5caC: 5-carboxylcytosine
KRuO4: potassium perruthenate